# Opportunities for Artificial Intelligence in Operational Medicine: Lessons from the United States Military

**DOI:** 10.3390/bioengineering12050519

**Published:** 2025-05-14

**Authors:** Nikolai Rakhilin, H. Douglas Morris, Dzung L. Pham, Maureen N. Hood, Vincent B. Ho

**Affiliations:** Department of Radiology and Bioengineering, Uniformed Services University for Health Science, 4301 Jones Bridge Rd, Bethesda, MD 20814, USA; herman.morris@usuhs.edu (H.D.M.); dzung.pham@usuhs.edu (D.L.P.); maureen.hood@usuhs.edu (M.N.H.); vincent.ho@usuhs.edu (V.B.H.)

**Keywords:** artificial intelligence, machine learning, biomedical technology, bioengineering, disaster medicine, military medicine

## Abstract

Conducted in challenging environments such as disaster or conflict areas, operational medicine presents unique challenges for the delivery of efficient and quality healthcare. It exposes first responders and medical personnel to many unexpected health risks and dangerous situations. To tackle these issues, artificial intelligence (AI) has been progressively incorporated into operational medicine, both on the front lines and also more recently in support roles. The ability of AI to rapidly analyze high-dimensional data and make inferences has opened up a wide variety of opportunities and increased efficiency for its early adopters, notably for the United States military, for non-invasive medical imaging and for mental health applications. This review discusses the current state of AI and highlights its broad array of potential applications in operational medicine as developed for the United States military.

## 1. Background

Artificial intelligence research is advancing at a rapid pace in conjunction with significant breakthroughs in the past few years that promise to revolutionize operational medicine. The AI field has seen substantial investment from and development by the United States (US) government and private industry, as exemplified by the USD 500 billion invested as part of the Project Stargate initiative in 2025 alone [1]. So far, AI systems have achieved remarkable capabilities, from being able to understand complex text (natural language processing, NLP) to interpreting unstructured visual inputs (computer vision) to using multimodal data to predict future events with high precision (predictive analytics) [2]. These capabilities have made them a vital part of the digital landscape, being integrated into robotics and AI assistants, such as Alexa or Siri, and enhancing our experience both on the internet and in daily life [3,4]. Tasks as complex as translating foreign text scribbled in a book can now be performed within seconds using AI on a portable device with a camera [5]. In the medical space, this also means that millions of pages of medical textbooks can be used to train an AI model that can prescribe medical treatments without provider input [6].

With the recent advancements in computational power and algorithms, AI has become capable of not only imitating human decision-making but also generating unique content (Generative AI) [7]. By leveraging multi-layered algorithms and statistical models, it can identify multi-dimensional patterns in data without human supervision and minimal guidance. Depending on the corpus it is exposed to, this truly disruptive technology can then create its own text, images, video, programming code, and music [8,9,10,11]. While such AI still falls short of human-level general intelligence, it excels at narrow, specialized tasks when large amounts of digital data are available to train on, which makes it perfectly suited for medical analysis, especially pertaining to operational medicine [12].

AI has slowly evolved from narrow algorithms to an indispensable tool integrated into numerous digital systems. It has become essential in the operational theater, where resources are limited and decisions need to be made rapidly. It is also incorporated into the lives of veterans, who need medical treatment after returning home. Here, we provide insights into these fields along with recommendations on how to be trained in using these new resources effectively.

## 2. Early AI

Early AI research focused on rule-based systems that laid the mathematical groundwork for large data analysis. Machine learning, which emerged in the 1950s, initially relied on statistical methods and simple algorithms to learn from data and generate predicted outcomes [13]. Widrow and Hoff applied these signal processing algorithms to create adaptive filters to isolate noise from communication systems [14]. As computational power increased and larger datasets became available, more complex models emerged. The concept of artificial neural networks, inspired by human brain architectures, was further expanded to allow for the recognition of patterns in images, demonstrated by Kohonen’s self-organizing map neural network, which could interpret map features without human guidelines (unsupervised learning) [15]. While remarkable for its time, this research had a limited impact due to the computational processing power constraints and digital data availability of the time [16].

A breakthrough was made by Hinton in the mid-2000s with the introduction of deep learning, characterized by neural networks with multiple hidden layers (hence “deep”) [17]. Each layer progressively extracts higher-level features from the input, identifying more abstract patterns the deeper it goes. This approach aimed to further replicate the higher-order learning mechanisms of human brains, which contain millions of neurons working in parallel to deconstruct complex ideas into abstract ones. The use of these layers led to rapid advancements in areas like computer vision, NLP, and speech recognition, leading to the development of specialized deep learning models like convolutional neural networks (CNNs) and recurrent neural networks (RNNs). CNNs use sliding filters to extract grid-like data across multiple layers, making them optimal for image processing, while RNNs can handle sequential data by maintaining an internal memory, making them powerful tools for language and audio analysis. AlexNet was one of the first models to implement this unique CNN architecture, with five convolutional and three connected layers, combined with the increasing performance of GPUs (graphics processing units) to make a giant leap forward in image recognition efficiency, drastically outperforming other traditional computer vision models and positioning GPUs as a key component for future AI hardware [18,19].

Another pivotal moment in the development of AI came in 2017 with the introduction of a transformer architecture, which revolutionized deep learning by providing self-attention mechanisms which can efficiently process relationships between different parts of the input data simultaneously [20]. This allowed for the faster analysis of high-resolution image and video data, enabling algorithms to implement billions of parameters simultaneously. This paved the way for the development of Large Language Models (LLMs) like Google’s BERT [21] in 2018 and OpenAI’s GPT-3 in 2020 [22], followed by ChatGPT-3 in 2022 [23], which were built on the principle of transformers. They demonstrated unprecedented language understanding and generation capabilities due to their scalable transformer architecture, increases in computational power, and massive digital repositories. The capabilities of AI are pushed further every year with more advancements, such as AI agents being incorporated into DeepSeek-R1 or the implementation of AI-specific hardware incorporated into NVIDIA GPUs (graphic processing units), drawing us closer to achieving the goal of artificial general intelligence (AGI) or superintelligence [24,25,26]. Thanks to these rapid developments and the widespread digitization of medical data, a diverse range of information (from sloppily written medical notes to high-resolution MRI scans) can now be efficiently incorporated into AI training datasets. The ability to process and learn from such varied data sources is increasingly being used to optimize patient care, streamline diagnostics, and support clinical decision-making.

## 3. AI Health Applications in Austere Locations: Lessons from the US Military

The military has long been an early adopter of technology for use in operational medicine. During the Civil War, the military introduced the use of the ambulance to transport wounded soldiers from the battlefield to treatment facilities [27]. This resulted in the creation of the first Ambulance Corps. Not surprisingly, the adoption of AI, including generative models, has also been a top priority for the US Department of Defense (DoD), which has had the responsibility of caring for 9.5 million patients [28]. This includes active-duty service members deployed globally, often to remote regions overseas, as well as those responding to natural disasters in North America or abroad.

AI tools have potential benefits for both disaster relief and combat casualty care in conflict zones (Figure 1). To take advantage of this technology, the US Army Futures Command set out a roadmap in 2022 on how it plans to have AI assist with decision-making in military medicine [29]. The roadmap included the use of AI-trained digital assistants to strategically prioritize patient treatments by analyzing extensive medical data, especially in scenarios with constrained resources or limited specialist availability. The DoD also sponsored a series of projects to develop novel AI-based technologies, many of which are summarized in Table 1. In one of these, the Defense Advanced Research Projects Agency (DARPA) sponsored the development of the University of Pittsburgh’s TRACIR (Trauma Care In a Rucksack) program [30]. The model was trained on over 7000 prehospital trauma patient datasets from the University of Pittsburgh Medical Center’s StatMedEvac medical service to analyze patient trauma and provide predictive outcomes. An additional project, titled “In The Moment (ITM)”, aims to meet similar goals by providing a virtual medic on any battlefield to help with the triage of wounded patients [31,32]. Similar AI medic models are also being developed to be able to detect early symptoms of shock [33], tension pneumothorax [34], and even hemorrhage/traumatic brain injury [35,36,37], giving on-site medics crucial specialized patient care advice quickly to enable prompt decision-making and improve patient outcomes. These AI tools can also provide critical information on the anticipated outcomes (typically up to 60 min into the future) to both facilitate proper triage on a single-patient level and streamline the distribution of critical resources in cases of mass casualty events.

These AI systems also help predict the need for hospitalization or emergency surgical procedures, such as amputations, during the “golden hour” (within 60 min of the injury). While often necessary, emergency field amputations show a four-fold increased risk of cardiovascular disease (for lower-limb amputations at or above the knee), an increased risk of severe lower back pain by 31%, and can cause post-traumatic stress disorder (PTSD), with 50–80% of patients report phantom pain [47]. AI can be used to mitigate many of these outcomes by assessing the need for immediate amputation [48] and predicting the limb revascularization recovery outcomes [49]. Since providing point-of-care treatments is not always possible, AI has also been used to optimize the medical evacuation (MEDEVAC) of wounded patients to enhance efficiency, location, and dispatching procedures in recent warzones, such as in Afghanistan [50]. This time-sensitive decision-making has been an explicit priority for the US DoD since 2009, leading to a drop in military fatality rates from 13.7% to 7.6% by 2014 [51].

Innovative AI approaches are being developed to enable the use of medical analytics, traditionally confined to hospitals, in operational medicine scenarios to improve patient triage and outcomes. The MySurgeryRisk model, developed by researchers at the University of Florida, is an AI model that was designed to predict the risks of several major postoperative complications, including sepsis, thromboembolism, and acute kidney injury, with 82–94% accuracy [52]. It is conceivable that AI tools, once developed and trained in hospital facilities in the US, could be adapted for and transferred for use in field hospitals or to more remote aid stations.

AI is also being implemented in the Medical Common Operating Picture (MedCOP), which is an interactive platform that provides real-time operational medical information and analysis [40]. Additionally, it is being integrated into wearable sensors for use by service members. These sensors record data on service members’ biometrics, environmental exposure, location and movement, and biomechanical stress [53]. This approach, known as biosurveillance, offers commanders a powerful tool to assess the health and readiness of their troops [54,55,56]. Such rapid, portable decision-making tools not only provide near real-time health monitoring, but also allow for smaller, more flexible deployments without the need for a large complement of highly trained healthcare specialists within medical units. Properly developed and implemented AI healthcare solutions provide military planners with flexibility in planning deployments and/or enhance the efficient use of medical resources, which in remote locations are often in short supply.

In military hospitals, AI can make a dramatic difference to clinical readiness and the early detection of potential risks for a medical disability. AI models can assess the current state of preparedness to identify weaknesses that can put service members at risk, resulting in both physical and mental injuries. One program, MITRE’s Medical Evaluation Readiness Information Toolset (MERIT) program [38], has contracts with the DoD to implement the prediction of the likelihood of service members entering the Disability Evaluation System (DES) within the next 6 months by analyzing digital military health data and correlating it with future outcomes. Other programs, such as MHS-GENESIS [44] and the Defense Innovation Unit’s Predictive Health algorithm [46], use AI to assist in military health screening by expanding the functionality of current medical software. These powerful programs can be digital canaries in the coalmine in identifying any potential medical issues in personnel and those undergoing training, ensuring troop readiness.

## 4. AI for Military Medical Imaging

Radiology plays a crucial role in providing critical, non-invasive medical data, especially in emergency medical and trauma settings. Over thirty years ago, the Digital Imaging and Communications in Medicine (DICOM) standard was adopted by Radiology Departments in the United States, including in the military, for the storage and exchange of digital medical images. The transition of medical imaging data from film to digital formats has enabled the development and application of AI tools. In addition to the DICOM standard, the DoD was also an early adopter of digital radiography and Picture Archiving and Communication System (PACS) technology, which facilitated its use of teleradiology [27]. Historically, this capability had been limited to major hospitals and low patient volumes due to the complexity of the equipment along with the need for specialized operators. AI-augmented radiology accelerates this process by facilitating or automating the acquisition and post-processing protocols, enabling the deployment of advanced radiology services in areas where they were otherwise unavailable.

Admittedly, the majority of AI development and use in military radiology has remained primarily in medical centers, similarly to that in civilian hospitals. For X-Ray Computed Tomography (CT), a fixed number of X-ray photons is typically required to pass through each subject to sufficiently resolve the internal structures. Standard CT images require 100× the amount of radiation used to produce a single X-ray image to produce a 3D volume [57]. While their high-resolution output provides critical medical insights, it comes at the cost of increased radiation exposure. Implementing AI reconstruction systems by training them on low- and high-resolution images can reduce the X-ray exposure needed to produce an equivalent CT volume image. AI-enabled reconstructions can then produce a complete anatomical image with a lower noise floor. This streamlines the clinical workflow for the limited number of radiologists on site [58,59,60]. For example, the military has developed a deployable CT scanner (Philips Healthcare, Best, Netherlands) with built-in AI guidance for improved CT workflows, accuracy of scanning, and timely diagnosis [61,62].

Compared to CT, Magnetic Resonance Imaging (MRI) does not cause any ionizing radiation exposure but needs significantly more time to produce an image, which limits the patient throughput. The use of AI-enabled MRI software (such as AUTOMAP) can reduce the image acquisition time, increase the resolution, and lower the noise floor in the resulting diagnostic image 1.5- to 4.5-fold, depending on the subject (Figure 2) [63]. This is particularly important in portable low-magnetic-field MRI scanners (Hyperfine Swoop, Hyperfine, Inc., Guilford, CT, USA), which has an integrated AI system that can perform these tasks with minimal external support [64]. This low-magnetic field MR scanner is relatively mobile and can be deployed in a military hospital with minimal staff and resources in order to provide critical anatomical information on injured service members. Such portable systems are becoming much more advanced and common in large part due to the contribution of AI data processing models that can generate high-value images within the portable scanner’s limitations [65,66,67,68,69].

AI is expanding the capabilities and enhancing the throughput of radiologists in more remote operational settings. The recent increased emphasis on developing teleradiology platforms allows for the natural integration of AI, enabling an individual radiologist to virtually support many medical units [70,71]. AI is particularly helpful when addressing medical cases that have no discernible pathology, as it can filter and pre-process much of the raw information. This allows both radiologists and sometimes non-radiologist medical personnel with less specialized training to assist in image interpretation [72,73].

## 5. AI for Mental Health

The mental health of service members is a critical concern in operational medicine. Military service members and emergency personnel are routinely exposed to traumatic events and the suffering of others, which can have profound psychological impacts on as they witness the severity of the pain and suffering of the wounded victims of natural disasters or conflicts, leading to a higher incidence of mental illness and suicide [74,75]. Service members are often reluctant to share their concerns and desire to harm themselves with their peers or their healthcare providers. Not surprisingly, service members and veterans are more willing to post their concerns in support groups, notably on social media platforms, that are not part of their work environment and outside of their usual healthcare system. The use of AI to evaluate social media posts by service members and veterans is a novel application of AI for the identification of those with suicidal ideation and to improve prevention [45]. Using their RoBERTa AI model, Zuromski was able to evaluate the social media posts of service members and military veterans to identify individuals with a high likelihood of having suicidal thoughts and behavior (sensitivity, 0.85; specificity, 0.96; precision, 0.64). The ability to use a third-party social media website presents an opportunity to identify a need for and provide early intervention, potentially saving the lives of service members and first responders who otherwise may not be known to have suicidal ideation.

AI can further provide assistance to veterans who have completed their service and are recovering. Recovery Engagement and Coordination for Health—Veterans Enhanced Treatment (REACH-VET) is an AI algorithm developed by the Department of Veterans Affairs (VA) in 2017 to evaluate the health of veterans who have returned home [39,76]. By examining veterans’ military health records, it can identify the top 0.1% of candidates who are most likely to be at risk for suicide, who can then be referred to VA coordinators who can help the veteran before it is too late, flagging 6700 veterans per month. Early intervention is key to the prevention of suicide, which is crucial since veterans are at a 57.3% higher risk of suicide than the general public, making it the second-leading cause of death among veterans [77]. 

Beyond suicide prevention, the use of AI is crucial to ensuring veterans’ wellbeing and rehabilitation in a civilian environment. To identify PTSD in the early stages of its development, New York University’s Langone Health researchers have developed an AI-based algorithm to detect PTSD using speech-based markers with 89% accuracy [78]. To further provide mental health support to veterans at home, the military has invested in the development of AI chatbots trained in the treatment of mental health disorders, such ReflexAI’s HomeTeam [41] and the USC ICT’s (University of Southern California Institute for Creative Technologies) Ellie [42], that can help provide 24/7 emergency counseling to veterans experiencing a mental health crisis. This allows for the provision of mental health support around the clock. These tools are becoming vital to combat the ongoing crisis of suicide in veterans, though they still require caution by users and oversight by mental health professionals.

As shown in the above examples, AI’s strongest asset is its ability to analyze vast amounts of data accurately and quickly, enabling rapid decision-making processes across multiple fields. Time-consuming tasks can be automated thanks to AI, and unlike past algorithms, it can be applied to more complex datasets including audio, video, and images. Furthermore, with increased amounts of data, AI systems can improve their performance and optimize their output, leading to further improvement. As such, since 2018, the DoD has invested in the JAIC (Joint Artificial Intelligence Center) to further accelerate the adoption of AI to improve missions’ efficacy [43]. With the new developments, these flexible systems can be used in the field, on base, and back home to help millions of service members make crucial medical decisions regardless of their location. 

## 6. Limitations of AI

Currently, AI still faces significant limitations in its use. Most crucially, an AI network is only as good as the dataset that it is trained on and the knowledge of the subject matter experts being consulted during the development of the algorithm. The training dataset, moreover, is created and selected by the consulting subject matter experts, and thus the collection and accurate annotation of data is potentially affected by human error. Biased, limited, or inconsistent datasets can lead AI to make poor decisions or overly simplified determinations in novel situations that are not part of the training set. A biased group of patients, skewed either by their race, age, or other characteristics, may also cause the AI network to draw inaccurate conclusions with a massive medical impact [79]. Deep learning models can potentially overcome these concerns by expanding their training database, such as how ChatGPT-3 is being trained on a massive dataset of 45 Terabytes of compressed plaintext [80], while the 15-year dataset in the US Joint Trauma System Department of Defense Trauma Registry (DoDTR) contains only 0.017 Gigabytes of data, despite including over 140,000 patient records [81,82]. Such a large dataset is difficult to obtain in large part due to legitimate concerns related to patient privacy and surveillance, as medical data HIPAA protections and privacy laws ensure that AI algorithms do not exploit people’s personal data for their training without consent.

To avoid infringing on personal data, large public (and private) databases have been established that can be used for AI training [83,84,85]. Additionally, the usage of information-dense imaging data and changing the method of data collection to automated digital inputs can minimize the impact of bias. However, even with a large dataset, implementing AI in a new, untested environment is always risky due to undetected bias, so rigorous testing is essential to ensure reliable results.

Once trained, the use of AI in the field may be limited by access to resources, such as power and internet connectivity. Despite recent rapid advances in AI hardware and models, most advanced models are optimized for high-power GPUs, which limits their portability or may mean they require an internet connection [86]. As such, developments in satellite internet connectivity have been critical in the distribution of personalized AI for service members [87].

The other main concern regarding AI is the “black box” nature of AI, which makes it challenging to follow the decision-making logic of AI systems, thus further reducing trust in the system. This can become a security liability, limiting its compliance with military communication and security requirements. While new approaches, like Explainable Artificial Intelligence (XAI) [88], are being implemented to allow for a clearer trail of logic, many AI systems often sacrifice clarity for accuracy and efficiency.

Lastly, while AI is helpful in generating medical advice and analyzing data, final patient care decisions require human input to ensure ethical implementation and quality control when dealing with medically critical events. As such, in 2020 the US Department of Defense (DoD) set out ethical principles for AI focused on five core pillars: governable, reliable, traceable, equitable, and responsible AI [89]. With NATO nations closely following suit [90], these principles will help if actively enforced and updated with the breakneck changes in AI development. Oversight, both system-wide and on a local device basis, ensures that these systems emulate the decision-making of military physicians and contribute to bettering medical care in the field [91].

## 7. Future of AI

Despite its limitations, AI will undoubtedly play a crucial role in operational medicine moving forward as the world becomes more digitized and data continues to be generated at increasingly fast rates. On the battlefield or in the aftermath of a natural disaster, AI has the potential to put medical knowledge in the palm of every service member or relief worker, rapidly imparting triage advice and performing diagnostics when outside assistance or the availability of more specialized healthcare personnel is limited. In hospitals, robotic surgeons are being trained to perform precise, repetitive tasks, while AI models are being trained on ultra-high-resolution medical imaging data to detect injuries and diseases to allow for early intervention. Post-deployment, AI is already being used for monitoring service member and veteran wellbeing, tracking a deluge of medical data from wearables, social media, and internet-of-things devices to identify medical complications before they occur.

In order for AI to be adopted in practice, it needs to be reliably integrated into the healthcare workforce of tomorrow. New medical students are now being trained on how AI functions and implementing AI in their work, with up to 50% of radiology practices saying that they already use or plan to use AI in their practices [92]. At the Uniformed Services University, medical residents are being exposed to AI technology to interpret X-rays and histopathology slides, not only making them familiar with the software, but teaching them its strengths and pitfalls [93]. Considering that the current amount of medical knowledge doubles every 75 days, leveraging AI will be crucial for physicians to maintain and improve the quality of care for service members [94].

## 8. How to Get Started in the World of AI

While designing large, cutting-edge LLMs requires teams of specialists, getting started in the world of AI is simpler than ever before. The military has also expressed interest in increasing their involvement in the digital space. For those without programming experience, testing out existing AI models can be a useful entry point to understand and test their limits. Chatbots employing AI, such as OpenAI’s ChatGPT, Microsoft’s Copilot, Google’s Gemini, and Perplexity’s Perplexity, allow users to generate advanced programs based on prompts [95]. This is taken a step further by text-to-image AIs, such as Canva’s AI Image Generator, Adobe’s Firefly, or DeepAI’s Text2Img, and then even further with the recent developments in text-to-video AIs, such as Runway’s GEN-2, Midjourney, and OpenAI’s Sora [96,97]. 

To get deeper into the programming aspect of AI, it is best to develop strong programming skills in Python, which is the most common language in AI development (Figure 3). Afterwards, obtain experience in machine learning-specific packages, such as TensorFlow, PyTorch, and scikit-learn. These packages will give you the tools needed to build your first machine learning platform and perform predictive analysis. To hone your skills in using these tools, Kaggle provides free repositories of datasets and pre-trained models for users to test their projects and collaborate using Kaggle Notebooks. Kaggle courses and challenges can help new users implement their programming skills and test their models.

Further learning can be achieved through a plethora of online courses and certificates offered by top universities and companies. Universities, such as Stanford University, the University of California, Berkeley, Carnegie Mellon University, Cornell University, and Columbia University, offer courses in beginner and advanced machine learning development [98,99,100,101,102,103,104]. This selection is further expanded by companies and non-profit organizations such as Google, Kaggle, Microsoft, and IBM, among others [105,106,107,108]. These courses and practice datasets can help both beginners and advanced users begin to design the next generation of AI systems and help promote the health of service members both at home and abroad [109]. 

## 9. Summary

Operational medicine involves healthcare delivery in resource-constrained environments, such as after major hurricanes and wildfires or in active warzones. In such instances, the rapid delivery of quality medical care is critical and requires speedy assessment and clinical decision-making. The ability of AI tools to provide expert medical knowledge, analysis, and advice will greatly improve the ability to care for the injured and allow healthcare workers to focus their efforts on those who require the most urgent care. Through the use of a multitude of tools either in development or already being implemented, the U.S. military has embraced AI for its application in healthcare and presents excellent use cases for civilian relief applications. Civilian relief organizations are assuredly not far behind in terms of their recognition of the improved efficiency that AI provides for operational medicine.

## Figures and Tables

**Figure 1 bioengineering-12-00519-f001:**
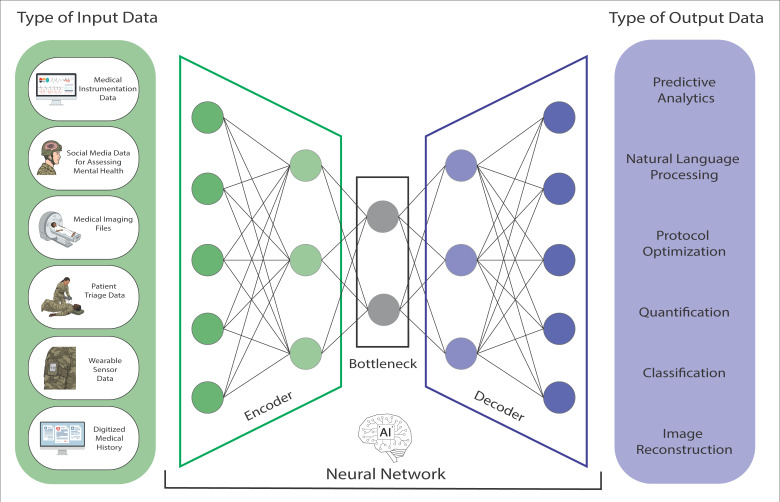
Visualization of a neural network used in operational medicine. Neural networks can use a variety of data types (**left** column) and process them across several layers, typically including an encoder, a bottleneck, and then a decoder, in which connected nodes are able to learn complex patterns. A trained network can then produce a wide array of outputs (**right** column) that can facilitate the completion of tasks related to operational medicine.

**Figure 2 bioengineering-12-00519-f002:**
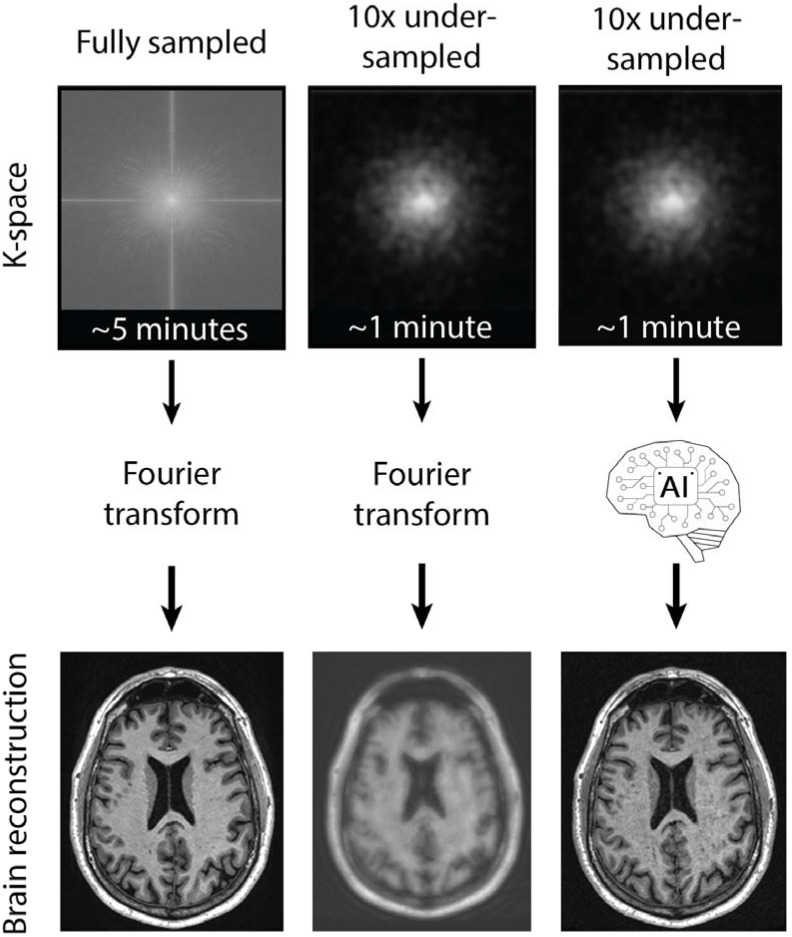
Accelerated acquisition of MRI data using AI. The traditional collection of MRI frequency data (k-space) takes a long time (**left**). Undersampling the frequency data can reduce the acquisition time but produces a low-resolution output (**middle**). AI is able to process undersampled data rapidly to reconstruct high-resolution data (**right**).

**Figure 3 bioengineering-12-00519-f003:**
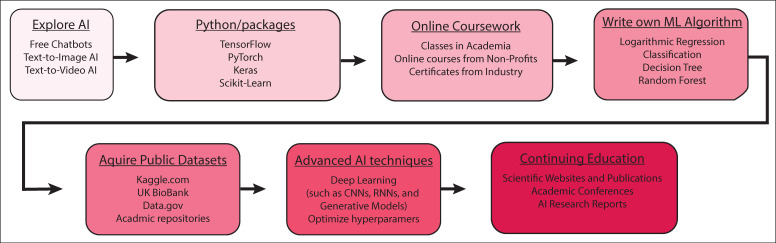
Roadmap for getting started in AI. To implement AI, one can start by exploring AI platforms, progressing to using essential Python packages, taking courses, composing an ML algorithm, incorporating data from publicly available datasets, processing data using more advanced AI techniques, and staying involved in recent AI developments through conferences and publications.

**Table 1 bioengineering-12-00519-t001:** Examples of AI systems that have either been implemented in operational medicine or are still in development.

Developer	Project Name	Features	Current Status
MITRE	MERIT [38]	Predictive modeling to identify service members at risk for disability	Implemented
Department of Veterans Affairs	REACH-VET [39]	Identification of veterans at risk for suicide achieved by analyzing health records	Implemented
Joint Health Services	MedCOP [40]	Data synchronization and real-time sharing of information from wearable sensors	Implemented
ReflexAI	HomeTeam [41]	Chatbot that provides emergency counseling, available 24 h per day	Implemented
USC Institute for Creative Technologies (DARPA-funded)	Ellie [42]	AI virtual therapist that assists in the diagnosis of mental illness and provides summaries for the provider	Implemented
DoD	JAIC [43]	Central hub of AI technologies to accelerate adoption and integration of AI in military medicine	Implemented
USMEPCOM	MHS GENESIS [44]	Uses AI to prescreen personnel for medical treatment	Implemented
Harvard University	RoBERTa [45]	Screening of social media posts to identify potential suicide ideation	Implemented
DARPA	ITM [31,32]	AI-integrated decision-making programs for battlefield triage	In Development
DoD Defense Innovation Unit	Predictive Health [46]	Uses AI to screen for cancers and other medical irregularities	In Development
University of Pittsburgh (DoD-sponsored)	TRACIR [30]	Provides autonomous trauma care and predictive analytics in remote locations	In Development

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
