# Peer review of "Opportunities for Artificial Intelligence in Operational Medicine: Lessons from the United States Military"

_bioengineering, 2025, doi:10.3390/bioengineering12050519_

Round 1
Reviewer 1 Report
Comments and Suggestions for Authors
This manuscript reviews the information about AI used in operational medicine. The section structure and writing skill of this review are appropriate. Therefore,this review is helpful for ones interested in the medicine applications of AI. But, no figures are used to explain the contents more clearly. Major revision with some figures is necessary.
Reviewer 2 Report
Comments and Suggestions for Authors
The review is devoted to the use of AI in military medicine. The main problem of the review is the phrase "military medicine". The review presents two real tasks facing AI: 1. Decoding medical data from X-rays and tomography. 2. Surveillance of people on the Internet. These are two well-known and solved problems from civilian medicine and law enforcement work. Perhaps in combat conditions someone uses several tomographs, perhaps there is a need to monitor veterans, perhaps, but it is doubtful! In general, I do not think that the title of the review matches the content.
AI works only with information, obviously, for this there must be sensors that are capable of receiving the necessary information. I do not know what sensors the US military wears, but comparing it with one of the most combat-ready armies of a potential enemy of the USA, I can assume that they wear a lot. Obviously, these sensors are capable of providing primary information for decision-making, but the review does not say a word about this!!! In all other cases, the doctor will have to obtain the information, then upload the information to the computer, then wait for the result. In general, it is not viable, at least for now.
The review does not mention computing equipment and computers. It is obvious that in combat conditions and even in the immediate rear, there is often no constantly available Internet. This allows us to say that doctors need to carry/have computing equipment for AI. In the current reality, this is a fairly large server consuming 3-4 kW of electricity. How do the authors imagine all this?
Half of the information that the authors rely on when writing the review is not strictly scientific and reliable. Often these are links to sites advertising certain commercial or semi-commercial products! Moreover, the review even contains direct advertising: "To get deeper into the programming aspect of AI, it is best to develop strong programming skills in Python, which is the most common language in AI development. Afterwards, getting experience in machine learning-specific packages, such as Tensor-Flow, PyTorch, and scikit-learn. These packages will give the tools needed to build your first machine learning platform and perform predictive analysis. To hone these tools, Kaggle provides vast, free repositories of datasets and pre-trained models for users to test their projects and collaborate with Kaggle Notebooks. Kaggle courses and challenges can help new users implement their programming skills and test their models».
The review contains non-informative illustrative materials. We will not discuss Table 2, it is an advertisement for educational institutions. Table 1 contains a very brief description of some examples of AI application, in which the first two columns are the developer and the commercial name. The prospects for using AI are also vague, by the way, this section could have drawn an excellent picture.
In general, the review is rather "raw", it is not clear who will read it and be educated. The authors need to seriously improve the quality!
Round 2
Reviewer 1 Report
Comments and Suggestions for Authors
This revised manuscript is ready to be published.
Reviewer 2 Report
Comments and Suggestions for Authors
I hope I helped the authors and I hope that the review will be useful to someone